# Stochastic Hyperparameter Optimization through Hypernetworks

## Abstract

Machine learning models are usually tuned by nesting optimization of model weights inside the optimization of hyperparameters. We give a method to collapse this nested optimization into joint stochastic optimization of both weights and hyperparameters. Our method trains a neural network to output approximately optimal weights as a function of hyperparameters. We show that our method converges to locally optimal weights and hyperparameters for sufficiently large hypernets. We compare this method to standard hyperparameter optimization strategies and demonstrate its effectiveness for tuning thousands of hyperparameters.

## 1 Introduction

Model selection and hyperparameter tuning is a major bottleneck in designing predictive models. Hyperparameter optimization can be seen as a nested optimization: The inner optimization finds model parameters $w$ which minimizes the training loss $\mathcal{L}_{\text{Train}}$ given hyperparameters $\lambda$. The outer optimization chooses $\lambda$ to minimize a validation loss $\mathcal{L}_{\text{Valid.}}$:

$$\operatorname*{argmin}_{\lambda} \mathcal{L}_{\text{Valid.}} \left( \operatorname*{argmin}_{w} \mathcal{L}_{\text{Train}} (w, \lambda) \right) \quad (1)$$

Standard practice in machine learning solves (1) by gradient-free optimization of hyperparameters, such as grid search, random search, or Bayesian optimization. Each set of hyperparameters is evaluated by re-initializing weights and training the model to completion. This is wasteful, since it trains the model from scratch each time, even if the hyperparameters change a small amount. Hyperband (Li et al., 2016) and freeze-thaw Bayesian optimization (Swersky et al., 2014) resume model training and do not waste this effort. Furthermore, gradient-free optimization scales poorly beyond 10 or 20 dimensions.

How can we avoid re-training from scratch each time? We usually estimate the parameters with stochastic optimization, but the true optimal parameters are a deterministic function of the hyperparameters $\lambda$:

$$w^*(\lambda) = \operatorname*{argmin}_{w} \mathcal{L}_{\text{Train}} (w, \lambda) \quad (2)$$

We propose to *learn this function*. Specifically, we train a neural network with inputs of hyperparameters, and outputs of an approximately optimal set of weights given the hyperparameters.

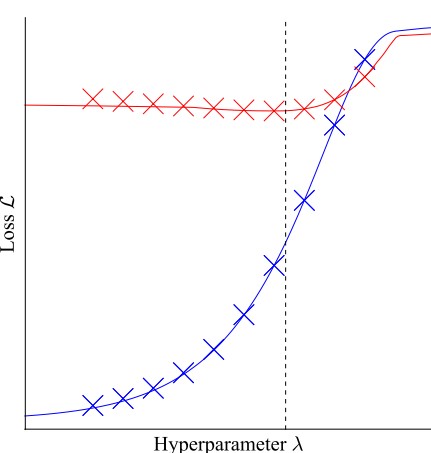

Train loss of optimized weights
Train loss of hypernet weights
Valid. loss of optimized weights
Valid. loss of hypernet weights
Optimal hyperparameter $\lambda$

Figure 1: Training and validation loss of a neural net, estimated by cross-validation (crosses) or by a hypernet (lines), which outputs $7,850$-dimensional network weights. The training and validation loss can be cheaply evaluated at any hyperparameter value using a hypernet. Standard cross-validation requires training from scratch each time.

This provides two major benefits: First, we can train the hypernet to convergence using stochastic gradient descent, denoted SGD, without training any particular model to completion. Second, differentiating through the hypernet allows us to optimize hyperparameters with gradient-based stochastic optimization.

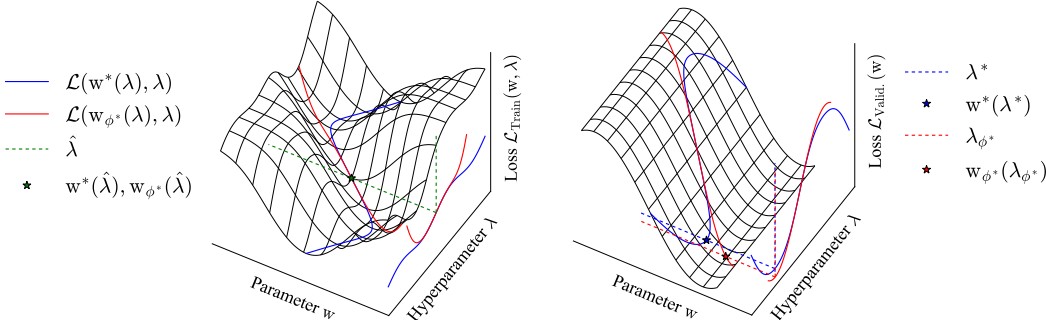

Figure 2: A visualization of exact (blue) and approximate (red) optimal weights as a function of given hyperparameters. *Left:* The training loss surface. *Right:* The validation loss surface. The approximately optimal weights $w_{\phi^*}$ are output by a linear model fit at $\hat{\lambda}$. The true optimal hyperparameter is $\lambda^*$, while the hyperparameter estimated using approximately optimal weights is nearby at $\lambda_{\phi^*}$.

## 2 TRAINING A NETWORK TO OUTPUT OPTIMAL WEIGHTS

How can we train a neural network to output approximately optimal weights of another neural network? A neural net which outputs the weights of another neural net is called a *hypernet* (Ha et al., 2016). The basic idea is that at each iteration, we ask a hypernet to output a set of weights given some hyperparameters: $w = w_\phi(\lambda)$. Instead of updating weights $w$ using the loss gradient $\partial\mathcal{L}(w)/\partial w$, we update the hypernet weights $\phi$ using the chain rule: $\frac{\partial\mathcal{L}(w_\phi)}{\partial w_\phi}\frac{\partial w_\phi}{\partial\phi}$. We call this method *hyper-training* and contrast it with standard training methods in Figure 3.

We call the function $w^*(\lambda)$ that outputs optimal weights for hyperparameters a *best-response function* (Fudenberg & Levine, 1998). At convergence, we want our hypernet $w_\phi(\lambda)$ to closely match the best-response function.

### 2.1 ADVANTAGES OF HYPERNET-BASED OPTIMIZATION

We can compare hyper-training to other model-based hyperparameter schemes, such as Bayesian optimization. Bayesian optimization (Snoek et al., 2012) builds a model of the validation loss as a function of hyperparameters, usually using a Gaussian process (Rasmussen & Williams, 2006) to track uncertainty. This approach has several disadvantages compared to hyper-training.

First, obtaining data for standard Bayesian optimization requires optimizing models from initialization for each set of hyperparameters. In contrast, hyper-training never needs to fully optimize any one model.

Second, standard Bayesian optimization treats the validation loss as a black-box function: $\hat{\mathcal{L}_{\text{Valid.}}}(\lambda) = f(\lambda)$. In contrast, hyper-training takes advantage of the fact that the validation loss is a known, differentiable function: $\hat{\mathcal{L}_{\text{Valid.}}}(\lambda) = \mathcal{L}_{\text{Valid.}}(w_\phi(\lambda))$ This removes the need to learn a model of the validation loss. This function can also be evaluated stochastically by sampling points from the validation set.

What sort of parameters can be optimized by our approach? Hyperparameters typically fall into two broad categories: 1) Optimization hyperparameters such as learning rates and initialization schemes, and 2) Regularization or model architecture parameters. Hyper-training *does not have inner optimization hyperparameters* because there is no inner training loop, so we can not optimize these. We must still choose optimization parameters for the fused optimization loop, but this is also the case for any model-based hyperparameter optimization method. The method can be applied to unconstrained bi-level optimization problems which consist of an inner loss function, inner parameter, outer loss function, and an outer parameter.

Algorithm 1: Standard cross-validation with stochastic optimization

1: **for** $i = 1, \ldots, T_{\text{outer}}$
2:     initialize w
3:     $\lambda = \text{hyperopt}\big(\ldots, \lambda^{(i)}, \mathcal{L}_{\text{Valid.}}\big(\text{w}^{(i)}\big)\big)$
4:     **for** $T_{\text{inner}}$ steps
5:         $\mathbf{x} \sim$ Training data
6:         $\text{w} = \text{w} - \alpha \nabla_{\text{w}} \mathcal{L}_{\text{Train}}(\mathbf{x}, \text{w}, \lambda)$
7:     $\lambda^i, \text{w}^i = \lambda, \text{w}$
8: **for** $i = 1, \ldots, T_{\text{outer}}$
9:     **if** $\mathcal{L}_{\text{Valid.}}\big(\text{w}^{(i)}\big) < \mathcal{L}_{\text{Valid.}}(\text{w})$ **then**
10:         $\hat{\lambda}, \text{w} = \lambda^i, \text{w}^i$
11: **return** $\hat{\lambda}, \text{w}$

Algorithm 2: Stochastic optimization of hypernet, then hyperparameters

1:
2: initialize $\phi$
3: initialize $\hat{\lambda}$
4: **for** $T_{\text{hypernet}}$ steps
5:     $\mathbf{x} \sim$ Training data, $\lambda \sim p(\lambda)$
6:     $\phi = \phi - \alpha \nabla_{\phi} \mathcal{L}_{\text{Train}}(\mathbf{x}, \text{w}_{\phi}(\lambda), \lambda)$
7:
8: **for** $T_{\text{hyperparameter}}$ steps
9:     $\mathbf{x} \sim$ Validation data
10:     $\hat{\lambda} = \hat{\lambda} - \beta \nabla_{\hat{\lambda}} \mathcal{L}_{\text{Valid.}}(\mathbf{x}, \text{w}_{\phi}(\hat{\lambda}))$
11: **return** $\hat{\lambda}, \text{w}_{\phi}(\hat{\lambda})$

Figure 3: A comparison of standard hyperparameter optimization, and our first algorithm. Instead of updating weights w using the loss gradient $\partial \mathcal{L}(\text{w})/\partial \text{w}$, we update hypernet weights $\phi$ using the chain rule: $\frac{\partial \mathcal{L}(\text{w}_{\phi})}{\partial \text{w}_{\phi}} \frac{\partial \text{w}_{\phi}}{\partial \phi}$. Also, instead of returning the best hyperparameters from a fixed set, our method uses gradient-based hyperparameter optimization. Here, hyperopt refers to a generic hyperparameter optimization.

## 2.2 LIMITATIONS OF HYPERNET-BASED OPTIMIZATION

Hyper-training can handle discrete hyperparameters but does not offer special advantages for optimizing over continuous hyperparameters. The hyperparameters being optimized must affect the training loss - this excludes optimization hyperparameters like learning rate. Also, our approach only proposes making local changes to the hyperparameters and does not do uncertainty-based exploration. Uncertainty can be incorporated into the hypernet by using stochastic variational inference as in Blundell et al. (2015), but we leave this for future work. Finally, it is not obvious how to choose the distribution over hyperparameters $p(\lambda)$. If we do not sample a sufficient number of hyperparameters we may reach sub-optimal solutions. We approach this problem in section 2.4.

An obvious difficulty of this approach is that training a hypernet typically requires training several times as many parameters as training a single model. For example, training a fully-connected hypernet with a single hidden layer of $H$ units to output $D$ parameters requires training at least $D \times H$ hypernet parameters. Again, in section 2.4 we propose an algorithm that requires training only a linear model mapping hyperparameters to model weights.

## 2.3 ASYMPTOTIC CONVERGENCE PROPERTIES

Algorithm 2 trains a hypernet using SGD, drawing hyperparameters from a fixed distribution $p(\lambda)$. This section proves that Algorithm 2 converges to a local best-response under mild assumptions. In particular, we show that, for a sufficiently large hypernet, the choice of $p(\lambda)$ does not matter as long as it has sufficient support. Notation as if $\text{w}_{\phi}$ has a unique solution for $\phi$ or w is used for simplicity, but is not true in general.

**Theorem 2.1.** *Sufficiently powerful hypernets can learn continuous best-response functions, which minimizes the expected loss for any hyperparameter distribution.*

$$\text{There exists } \phi^*, \text{ such that for all } \lambda \in \text{support}(p(\lambda)),$$
$$\mathcal{L}_{\text{Train}}(\text{w}_{\phi^*}(\lambda), \lambda) = \min_{\text{w}} \mathcal{L}_{\text{Train}}(\text{w}, \lambda)$$
$$\text{and } \phi^* = \operatorname*{argmin}_{\phi} \mathbb{E}_{p(\lambda')}\left[ \mathcal{L}_{\text{Train}}(\text{w}_{\phi}(\lambda'), \lambda') \right]$$

*Proof.* If $\text{w}_{\phi}$ is a universal approximator (Hornik, 1991) and the best-response is continuous in $\lambda$ (which allows approximation by $\text{w}_{\phi}$), then there exists optimal hypernet parameters $\phi^*$ such that for all hyperparameters $\lambda$, $\text{w}_{\phi^*}(\lambda) = \operatorname{argmin}_{\text{w}} \mathcal{L}_{\text{Train}}(\text{w}, \lambda)$. Thus, $\mathcal{L}_{\text{Train}}(\text{w}_{\phi^*}(\lambda), \lambda) = \min_{\text{w}} \mathcal{L}_{\text{Train}}(\text{w}, \lambda)$. In other words, universal approximator hypernets can learn continuous best-responses.

Algorithm 2: Stochastic optimization of hypernet, then hyperparameters

1: initialize $\phi, \hat{\lambda}$
2: **for** $T_{\text{hypernet}}$ steps
3:     $\mathbf{x} \sim$ Training data, $\lambda \sim p(\lambda)$
4:     $\phi = \phi - \alpha \nabla_\phi \mathcal{L}_{\text{Train}}(\mathbf{x}, w_\phi(\hat{\lambda}), \hat{\lambda})$
5: **for** $T_{\text{hyperparameter}}$ steps
6:     $\mathbf{x} \sim$ Validation data
7:     $\hat{\lambda} = \hat{\lambda} - \beta \nabla_{\hat{\lambda}} \mathcal{L}_{\text{Valid.}}(\mathbf{x}, w_\phi(\hat{\lambda}))$
8: **return** $\hat{\lambda}, w_\phi(\hat{\lambda})$

Algorithm 3: Stochastic optimization of hypernet and hyperparameters jointly

1: initialize $\phi, \hat{\lambda}$
2: **for** $T_{\text{joint}}$ steps
3:     $\mathbf{x} \sim$ Training data, $\lambda \sim p(\lambda|\hat{\lambda})$
4:     $\phi = \phi - \alpha \nabla_\phi \mathcal{L}_{\text{Train}}(\mathbf{x}, w_\phi(\hat{\lambda}), \hat{\lambda})$
5:
6:     $\mathbf{x} \sim$ Validation data
7:     $\hat{\lambda} = \hat{\lambda} - \beta \nabla_{\hat{\lambda}} \mathcal{L}_{\text{Valid.}}(\mathbf{x}, w_\phi(\hat{\lambda}))$
8: **return** $\hat{\lambda}, w_\phi(\hat{\lambda})$

Figure 4: A side-by-side comparison of two variants of hyper-training. Algorithm 3 fuses the hypernet training and hyperparameter optimization into a single loop of SGD.

Substituting $\phi^*$ into the training loss gives $\mathbb{E}_{p(\lambda)}[\mathcal{L}_{\text{Train}}(w_{\phi^*}(\lambda), \lambda)] = \mathbb{E}_{p(\lambda)}[\min_\phi \mathcal{L}_{\text{Train}}(w_\phi(\lambda), \lambda)]$. By Jensen's inequality, $\min_\phi \mathbb{E}_{p(\lambda)}[\mathcal{L}_{\text{Train}}(w_\phi(\lambda), \lambda)] \geq \mathbb{E}_{p(\lambda)}[\min_\phi \mathcal{L}_{\text{Train}}(w_\phi(\lambda), \lambda)]$ where $\min_\phi$ is a convex function on the convex vector space of functions $\{\mathcal{L}_{\text{Train}}(w_\phi(\lambda), \lambda) \text{ for } \lambda \in \text{support}(p(\lambda))\}$ if $\text{support}(p(\lambda))$ is convex and $\mathcal{L}_{\text{Train}}(w, \lambda) = \mathbb{E}_{\mathbf{x} \sim \text{Train}}[\mathcal{L}_{\text{Pred}}(\mathbf{x}, w)] + \mathcal{L}_{\text{Reg}}(w, \lambda)$ with $\mathcal{L}_{\text{Reg}}(w, \lambda) = \lambda \cdot \mathcal{L}(w)$. Thus, $\phi^* = \text{argmin}_\phi \mathbb{E}_{p(\lambda)}[\mathcal{L}_{\text{Train}}(w_\phi(\lambda), \lambda)]$. In other words, if the hypernet learns the best-response it will simultaneously minimize the loss for every point in the $\text{support}(p(\lambda))$. $\qquad\square$

Thus, having a universal approximator and a continuous best-response implies for all $\lambda \in \text{support}(p(\lambda))$, $\mathcal{L}_{\text{Valid.}}(w_{\phi^*}(\lambda)) = \mathcal{L}_{\text{Valid.}}(w^*(\lambda))$ because $w_{\phi^*}(\lambda) = w^*(\lambda)$. Thus, under mild conditions, we will learn a best-response in the support of the hyperparameter distribution.If the best-response is differentiable, then there is a neighborhood about each hyperparameter where the best-response is approximately linear. If we select the support of the hyperparameter distribution to be the neighborhood where the best-response is approximately linear then we can use linear regression to learn the best-response locally.

Theorem 2.1 holds for any $p(\lambda)$. However in practice, we have a limited-capacity hypernet, and so should choose a $p(\lambda)$ that puts most of its mass on promising hyperparameter values. This motivates the joint optimization of $\phi$ and $p(\lambda)$. Concretely, we can introduce a "current" hyperparameter $\hat{\lambda}$ and define a conditional hyperparameter distribution $p(\lambda|\hat{\lambda})$ which places its mass near $\hat{\lambda}$. This allows us to use a limited-capacity hypernet, at the cost of having to re-train the hypernet each time we update $\hat{\lambda}$.

In practice, there are no guarantees about the network being a universal approximator or the finite-time convergence of optimization. The optimal hypernet will depend on the hyperparameter distribution $p(\lambda)$, not just the support of this distribution. We appeal to experimental results that our method is feasible in practice.

## 2.4 JOINTLY TRAINING PARAMETERS AND HYPERPARAMETERS

Because in practice we use a limited-capacity hypernet, it may not be possible to learn a best-response for all hyperparameters. Thus, we propose Algorithm 3, which only tries to learn a best-response locally. We introduce a "current" hyperparameter $\hat{\lambda}$, which is updated each iteration. We define a conditional hyperparameter distribution, $p(\lambda|\hat{\lambda})$, which only puts mass close to $\hat{\lambda}$.

Algorithm 3 combines the two phases of Algorithm 2 into one. Instead of first learning a hypernet that can output weights for any hyperparameter then optimizing the hyperparameters, Algorithm 3 only samples hyperparameters near the current best guess. This means the hypernet only has to be trained to estimate good enough weights for a small set of hyperparameters. The locally-trained hypernet can then be used to provide gradients to update the hyperparameters based on validation set performance.

How simple can we make the hypernet, and still obtain useful gradients to optimize hyperparameters? Consider the case used in our experiments where the hypernet is a linear function of the hyperparameters and the conditional hyperparameter distribution is $p(\lambda|\hat{\lambda}) = \mathcal{N}(\hat{\lambda}, \sigma\mathbb{1})$ for some small $\sigma$. This hypernet learns a tangent hyperplane to a best-response function and only needs to make small adjustments at each step if the hyperparameter updates are sufficiently

small. We can further restrict the capacity of a linear hypernet by factorizing its weights, effectively adding a bottleneck layer with a linear activation and a small number of hidden units.

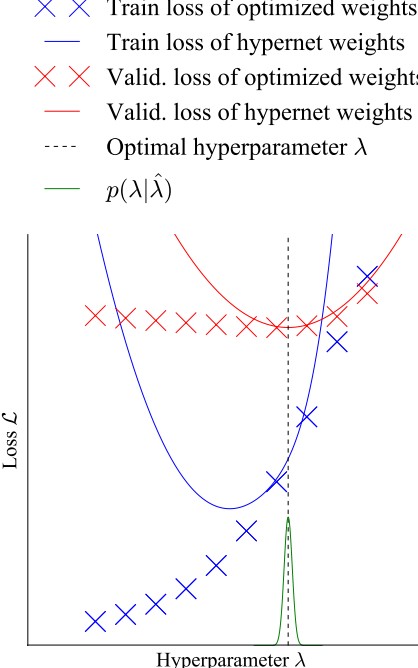

Figure 5: The training and validation losses of a neural network, estimated by cross-validation (crosses) or a linear hypernet (lines). The hypernet's limited capacity makes it only accurate where hyperparameter distribution put mass.

## 3 RELATED WORK

Our work is closely related to the concurrent work of Brock et al. (2017), whose SMASH algorithm also approximates the optimal weights as a function of model architectures, to perform a gradient-free search over discrete model structures. Their work focuses on efficiently evaluating the performance of a variety of discrete model architectures, while we focus on efficiently exploring continuous spaces of models.

**Model-free approaches** Model-free approaches only use trial-and-error to explore the hyperparameter space. Simple model-free approaches applied to hyperparameter optimization include grid search and random search (Bergstra & Bengio, 2012). Hyperband (Li et al., 2016) combines bandit approaches with modeling the learning procedure.

**Model-based approaches** Model-based approaches try to build a surrogate function, which often allows gradient-based optimization or active learning. A common example is Bayesian optimization (Snoek et al., 2012). Freeze-thaw Bayesian optimization can condition on partially-optimized model performance.

**Optimization-based approaches** Another line of related work attempts to directly approximate gradients of the validation loss with respect to hyperparameters. Domke (2012) proposes to differentiate through unrolled optimization to approximate best-responses in nested optimization and Maclaurin et al. (2015a) differentiate through entire unrolled learning procedures. DrMAD (Fu et al., 2016) approximates differentiating through an unrolled learning procedure to relax memory requirements for deep neural networks. HOAG (Pedregosa, 2016) finds hyperparameter gradients with implicit differentiation by deriving an implicit equation for the gradient with optimality conditions. Franceschi et al. (2017) study forward and reverse-mode differentiation for constructing hyperparameter gradients. Also, Feng & Simon (2017) establish conditions where the validation loss of best-responding weights are almost everywhere smooth, allowing gradient-based training of hyperparameters.

A closely-related procedure to our method is the $T1 - T2$ method of Luketina et al. (2016), which also provides an algorithm for stochastic gradient-based optimization of hyperparameters. The convergence of their procedure to local optima of the validation loss depends on approximating the Hessian of the training loss with respect to parameters with the identity matrix. In contrast, the convergence of our method depends on having a suitably powerful hypernet.

**Game theory** Best-response functions are extensively studied as a solution concept in discrete and continuous multi-agent games (Fudenberg & Levine, 1998). Games where learning a best-response can be applied include adversarial training (Goodfellow et al., 2014), or Stackelberg competitions (Brückner & Scheffer, 2011). For adversarial training, the analog of our method would be a discriminator which is trained while observing all of the generator's parameters.

## 4 EXPERIMENTS

In our experiments, we examine the standard example of stochastic gradient-based optimization of neural networks, with a weight regularization penalty. Gradient-based methods explicitly use

the gradient of a loss, while gradient-free methods do not (but can use the gradient of a surrogate loss that is learned). Our algorithm may be contrasted with gradient-free techniques like Bayesian optimization, gradient-based methods only handle hyperparameters affecting the training loss and gradient-based methods which can additionally handle optimization parameters. The best comparison for hyper-training is to gradient-based methods which only handle parameters affecting the training loss because other methods apply to a more general set of problems. In this case, the training and validation losses can be written as:

$$\mathcal{L}_{\text{Train}}(w, \lambda) = \mathop{\mathbb{E}}_{x \sim \text{Train}}\left[\mathcal{L}_{\text{Pred}}(\mathbf{x}, w)\right] + \mathcal{L}_{\text{Reg}}(w, \lambda)$$

$$\mathcal{L}_{\text{Valid.}}(w) = \mathop{\mathbb{E}}_{x \sim \text{Valid.}}\left[\mathcal{L}_{\text{Pred}}(\mathbf{x}, w)\right]$$

In all experiments, Algorithms 2 or 3 are used to optimize weights of with mean squared error on MNIST (LeCun et al., 1998) with $\mathcal{L}_{\text{Reg}}$ as an $L_2$ weight decay penalty weighted by $\exp(\lambda)$. The elementary model has $7,850$ weights. All hidden units in the hypernet have a ReLU activation (Nair & Hinton, 2010) unless otherwise specified. Autograd (Maclaurin et al., 2015b) was used to compute all derivatives. For each experiment, the minibatch samples 2 pairs of hyperparameters and up to $1,000$ training data points. Adam was used for training the hypernet and hyperparameters, with a step size of 0.0001. All experiments were run on a 2012 MacBook pro.

### 4.1 LEARNING A GLOBAL BEST-RESPONSE

Our first experiment, shown in Figure 1, demonstrates learning a global approximation to a best-response function using Algorithm 2. To make visualization of the regularization loss easier, we use 10 training data points to exacerbate overfitting. We compare the performance of weights output by the hypernet to those trained by standard cross-validation (Algorithm 1). Thus, elementary weights were randomly initialized for each hyperparameter setting and optimized using Adam (Kingma & Ba, 2014) for $1,000$ iterations with a step size of 0.0001.

When training the hypernetwork, hyperparameters were sampled from a broad Gaussian distribution: $p(\lambda) = \mathcal{N}(0, 1.5)$. The hypernet has 50 hidden units which results in $400,450$ parameters of the hypernetwork.

The minimum of the best-response in Figure 1 is close to the true minimum of the validation loss, which shows a hypernet can satisfactorily approximate a global best-response function in small problems.

### 4.2 LEARNING A LOCAL BEST-RESPONSE

Figure 5 shows the same experiment, but using the fused updates of Algorithm 3. This results in finding a best-response approximation whose minimum is the true minimum faster than the prior experiment. The conditional hyperparameter distribution is given by $p(\lambda|\hat{\lambda}) = \mathcal{N}(\hat{\lambda}, 0.00001)$. The hypernet is a linear model, with only $15,700$ weights. We use the same optimizer as the global best-response to update both the hypernet and the hyperparameters.

Again, the minimum of the best-response at the end of training is the true optimum on the validation loss. This experiment shows that using only a locally-trained linear best-response function can give sufficient gradient information to optimize hyperparameters on a small problem. Algorithm 3 is also less computationally expensive than Algorithms 1 or 2.

### 4.3 HYPER-TRAINING AND UNROLLED OPTIMIZATION

In order to compare hyper-training with other gradient-based hyperparameter optimization we trained a model with $7,850$ hyperparameters with a separate $L_2$ weight decay applied to each weight in a 1 layer (linear) model. The conditional hyperparameter distribution and optimizer for the hypernet and hyperparameters is the same the prior experiment. The weights for the model are factorized by selecting a hypernet with 10 hidden units. The factorized linear hypernet has 10 hidden units giving $164,860$ weights. This means each hypernet iteration $2 \cdot 10$ times as expensive as an iteration on just the model, because there is the same number of hyperparameters as model parameters.

Figure 6: Validation and test losses during hyperparameter optimization with a separate $L_2$ weight decay applied to each weight in the model. Thus, models with more parameters have more hyperparameters. *Left:* The $7,850$ dimensional hyperparameter optimization problem from having a linear model is solved with multiple algorithms. Hypernetwork-based optimization converges faster than unrolled optimization from Maclaurin et al. (2015a) but to a sub-optimal solution. *Right:* Hyper-training is applied different layer configurations in the model. The hand-tuned regularization parameters on the 784-10, 784-100-10, and 784-100-100-10 models have a validation losses of $0.434, 0.157$ and $0.206$ respectively.

Figure 6, left, shows that Algorithm 3 converges more quickly than the unrolled reverse-mode optimization introduced in Maclaurin et al. (2015a) with an implementation by Franceschi et al. (2017). Hyper-training reaches sub-optimal solutions because of limitations on how many hyperparameters can be sampled for each update but overfits validation data less than unrolling. Standard Bayesian optimization cannot be scaled to this many hyperparameters. Thus, this experiment shows Algorithm 3 can effectively partially optimize thousands of hyperparameters.

## 4.4 OPTIMIZING WITH DEEPER NETWORKS

In order to to see if we can optimize deeper networks with hyper-training we optimize models with 1, 2, and 3 layers with a separate $L_2$ weight decay applied to each weight. The conditional hyperparameter distribution and optimizer for the hypernet and hyperparameters is the same the prior experiment. The weights for each model are factorized by selecting a hypernet with 10 hidden units. Again, standard Bayesian optimization cannot be scaled to this many hyperparameters.

Figure 6, right, shows that Algorithm 3 can scale to networks with multiple hidden layers and outperform hand-tuned settings. As more layers are added the difference between validation loss and testing loss decreases, and the model performs better on the validation set. Future work should compare other architectures like recurrent or convolutional networks. Additionally, note that more layers perform with lower training (not shown), validation, and test losses, as opposed to lower training loss but higher validation or test loss. This indicates that using weight decay on each weight could be a prior for generalization, or that hyper-training enforces another useful prior like the continuity of a best-response.

## 4.5 ESTIMATING WEIGHTS VERSUS ESTIMATING LOSS

As mentioned above, our approach differs from Bayesian optimization in that we try to learn to predict optimal weights, while Bayesian optimization attempts to directly model the validation loss of optimized weights. In this final experiment, we untangle the reason for the better performance of our method: Is it because of a better inductive bias, or because our method can see more hyperparameter settings during optimization?

First, we constructed a hyper-training set: We optimized 25 sets of weights to completion, given randomly-sampled hyperparameters. We chose 25 samples since that is the regime in which we expect Gaussian process-based approaches to have the largest advantage. We also constructed a validation set of $10,215$ (optimized weight, hyperparameter) tuples generated in the same manner. We then fit a Gaussian process (GP) regression model with an RBF kernel from sklearn on the hyper-training data. A hypernet is fit to the same dataset. However, this hypernet was trained to

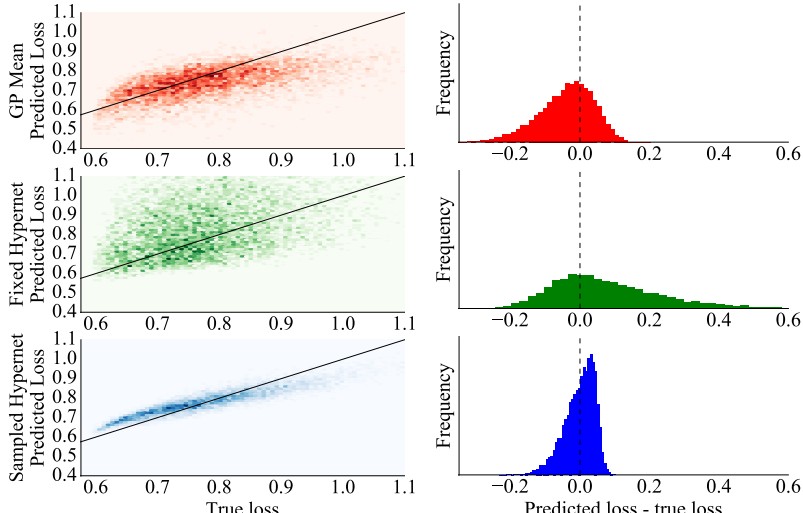

Figure 7: Comparing three approaches to predicting validation loss. *First row:* A Gaussian process, fit on a small set of hyperparameters and the corresponding validation losses. *Second row:* A hypernet, fit on the same small set of hyperparameters and the corresponding optimized weights. *Third row:* Our proposed method, a hypernet trained with stochastically sampled hyperparameters. *Left:* The distribution of predicted and true losses. The diagonal black line is where predicted loss equals true loss. *Right:* The distribution of differences between predicted and true losses. The Gaussian process often under-predicts the true loss, while the hypernet trained on the same data tends to over-predict the true loss.

fit optimized training weights, not optimized validation loss. Finally, we optimize another hypernet using Algorithm 2, for the same amount of time as building the hyper-training set. The two hypernets were linear models and were trained with the same optimizer parameters as the $7,850$-dimensional hyperparameter optimization.

Figure 7 shows the distribution of prediction errors of these three models. We can see that the Gaussian process tends to underestimate loss. The hypernet trained with the same small fixed set of examples tends to overestimate loss. We conjecture that this is due to the hypernetwork producing bad weights in regions where it doesn't have enough training data. Because the hypernet must provide actual weights to predict the validation loss, poorly-fit regions will overestimate the validation loss. Finally, the hypernet trained with Algorithm 2 produces errors tightly centered around 0. The main takeaway from this experiment is a hypernet can learn more accurate surrogate functions than a GP for equal compute budgets because it views (noisy) evaluations of more points.

Code for all experiments will be made available upon publication.

## 5 Conclusions

In this paper, we:

- Presented algorithms that efficiently learn a differentiable approximation to a best-response without nested optimization.

- Showed empirically that hypernets can provide a better inductive bias for hyperparameter optimization than Gaussian processes fit directly to the validation loss.

- Gave a theoretical justification that sufficiently large networks will learn the best-response for all hyperparameters it is trained against.

We hope that this initial exploration of stochastic hyperparameter optimization will inspire further refinements, such as hyper-regularization methods, or uncertainty-aware exploration using Bayesian hypernetworks.

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

## A    EXTRA EXPERIMENTS

### A.1    OPTIMIZING 10 HYPERPARAMETERS

Here, we optimize a model with 10 hyperparameters, in which a separate $L_2$ weight decay is applied to the weights for each digit class in a linear regression model to see if we can optimize medium-sized models. The conditional hyperparameter distribution and optimizer for the hypernet and hyperparameters is the same the prior experiments. A linear hypernet is used, resulting in $86,350$ hyper-weights. Algorithm 3 is compared against random search and .

Figure 8, right, shows that our method converges more quickly and to a better optimum than either alternative method, demonstrating that medium-sized hyperparameter optimization problems can be solved with Algorithm 3.

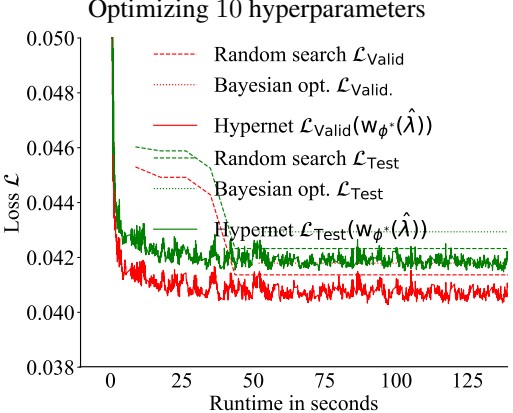

Figure 8: Validation and test losses during hyperparameter optimization. A separate $L_2$ weight decay is applied to the weights of each digit class, resulting in 10 hyperparameters. The weights $w_{\phi^*}$ are output by the hypernet for current hyperparameter $\lambda$, while random losses are for the best result of a random search. Hypernetwork-based optimization converges faster than random search or Bayesian optimization. We also observe significant overfitting of the hyperparameters on the validation set, which may be reduced be introducing hyperhyperparameters (parameters of the hyperparameter prior). The runtime includes the inner optimization for gradient-free approaches so that equal cumulative computational time is compared for each method.

Factors affecting this include removing the overhead of constructing tuples of hyperparameters and optimized weights, viewing more hyperparameter samples, or having a better inductive bias from learning weights.

