# OpenReview forum: "Stochastic Hyperparameter Optimization through Hypernetworks"
_ICLR.cc/2018/Conference — Reject_

### Official Review · AnonReviewer3 · 2017-11-24
**An interesting idea to optimize NN hyper-parameters questioned by the practical difficulties of addressing large models.**

**Rating:** 6
**Confidence:** 3

**Review:**

This paper introduces the use of hyper-networks for hyper-parameter optimization in the context of neural networks. A hyper-network is a network that has been trained to find optimal weights for another neural network on a particular learning task. This hyper-network can also be trained using gradient descent, and then can be optimized with respect to its inputs (hyper-parameters) to find optimal hyper-parameters. Of course, for this to be feasible training the hyper-network has to be efficient. For this, the authors suggest to use a linear hyper-network. The use of this approach for hyper-parameter optimization is illustrated in several experiments considering a linear model on the MNIST dataset.

The paper is clearly written with only a few typing errors.

As far as I know this work is original. This is the first time that hyper-networks are used for hyper-parameter optimization.

The significance of the work can, however, be questioned. To begin with, the models considered by the authors are rather small. They are simply linear models in which the number of weights is not very big. In particular, only 7,850 weights. The corresponding hyper-net has around 15,000, which is twice as big. Furthermore, the authors say that they train the hyper-network 10 times more than standard gradient descent on the hyper-parameter. This accounts for training the original model 20 times more.

If the original model is a deep neural network with several hidden layers and several hidden units in each layer, it is not clear if the proposed approach will be feasible. That is my main concern with this paper. The lack of representative models such as the ones used in practical applications.

Another limitation is that the proposed approach seems to be limited to neural network models. The other techniques the authors compare with are more general and can optimize the hyper-parameters of other models.

Something strange is that the authors claim in Figure 6 that the proposed method is able to optimize 7,850 hyper-parameters. However, it is not clear to what extent this is true. To begin with, it seems that the performance obtained is worse than with 10 hyper-parameters (shown on the right). Since the left case is a general case of the right case (having only 10 hyper-parameters different) it is strange that worse results are obtained in the left case. It seems that the optimization process is reaching sub-optimal solutions.

The experiments shown in Figure 7 are strange. I have not fully understood their importance or what conclusions can the authors extract from them.

I have also missed comparing with related techniques such as Dougal Maclaurin et al., 2015.

Summing up, this seems to be an interesting paper proposing an interesting idea. However, it seems the practical utility of the method described is limited to small models only, which questions the overall significance.

---

> ### Author Response · Authors · 2018-01-05
> **Response to Reviewer 3**
>
> Thank you for your detailed comments. I will address your comments in the order in which you wrote them. For clarity, I have placed your comments in [brackets] and my responses will follow in plain text.
>
> [The significance of the work can, however, be questioned. To begin with, the models considered by the authors are rather small. They are simply linear models in which the number of weights is not very big. In particular, only 7,850 weights. The corresponding hyper-net has around 15,000, which is twice as big. Furthermore, the authors say that they train the hyper-network 10 times more than standard gradient descent on the hyper-parameter. This accounts for training the original model 20 times more. ]
>
> We have added graphs showing performance on deeper models with more weights.  It is a good point that each iteration of hypernet is more expensive than an iteration of optimizing the elementary model.  For a linear hypernet it is about (# hyperparameters * # parameters) / (# parameters) = # hyperparameters times more expensive.  For our experiment where # hyperparameters = # parameters with 10 hidden units it is (# hyperparameters * 10 + 10 * # parameters) / (# parameters) = 20 times more expensive.
>
>
> [If the original model is a deep neural network with several hidden layers and several hidden units in each layer, it is not clear if the proposed approach will be feasible. That is my main concern with this paper. The lack of representative models such as the ones used in practical applications.]
>
>
> We have added comparisons to MLP with up with several hidden layers and several hidden units in each layer.  Further experiments should still be conducted on convolutional or recurrent networks.
>
>
> [Another limitation is that the proposed approach seems to be limited to neural network models. The other techniques the authors compare with are more general and can optimize the hyper-parameters of other models.]
>
>
> A good point is raised in that there are hyperparameters this algorithm can not optimize.  We can not optimize hyperparameters about optimization, because there is no inner optimization loop.  These methods can be applied to any (unconstrained) bi-level optimization, where we learn the inner parameters best-response to the outer-parameter with a neural network.  The method is applied to a specific case often encountered in machine learning where the inner parameter is model weights, while the outer parameter is hyperparameters.  SMASH uses a similar algorithm to do architecture search of neural networks.
>
>
> [Something strange is that the authors claim in Figure 6 that the proposed method is able to optimize 7,850 hyper-parameters. However, it is not clear to what extent this is true. To begin with, it seems that the performance obtained is worse than with 10 hyper-parameters (shown on the right). Since the left case is a general case of the right case (having only 10 hyper-parameters different) it is strange that worse results are obtained in the left case. It seems that the optimization process is reaching sub-optimal solutions.]
>
>
> This is a good point!  We have emphasized that the algorithm is reaching sub-optimal solutions due to limited capacity and sampling an insufficient number of hyperparameters on each iteration.  This is shown in the new experiment where we compare our algorithm to differentiating through optimization, which is slower but finds better solutions.
>
>
> [The experiments shown in Figure 7 are strange. I have not fully understood their importance or what conclusions can the authors extract from them.]
>
> The main takeaway from this experiment is a hypernet can learn more accurate surrogate functions than a GP for equal compute budgets because it views (noisy) evaluations of more points.  This has been added to the experiment
>
>
>
> [I have also missed comparing with related techniques such as Dougal Maclaurin et al., 2015.]
>
> We have added a comparison with differentiating through optimization from Dougal Maclaurin et al., 2015.

---

### Official Review · AnonReviewer1 · 2017-11-27
**Interesting idea to apply hyper-networks to tune hyper-parameters; however, paper lacks clarity, has technical concerns and experiments appear as artificial.**

**Rating:** 6
**Confidence:** 4

**Review:**

*Summary*

The paper proposes to use hyper-networks [Ha et al. 2016] for the tuning of hyper-parameters, along the lines of [Brock et al. 2017]. The core idea is to have a side neural network sufficiently expressive to learn the (large-scale, matrix-valued) mapping from a given configuration of hyper-parameters to the weights of the model we wish to tune.
The paper gives a theoretical justification of its approach, and then describes several variants of its core algorithm which mix the training of the hyper-networks together with the optimization of the hyper-parameters themselves. Finally, experiments based on MNIST illustrate the properties of the proposed approach.

While the core idea may appear as appealing, the paper suffers from several flaws (as further detailed afterwards):
-Insufficient related work
-Correctness/rigor of Theorem 2.1
-Clarity of the paper (e.g., Sec. 2.4)
-Experiments look somewhat artificial
-How scalable is the proposed approach in the perspective of tuning models way larger/more complex than those treated in the experiments?

*Detailed comments*

-"...and training the model to completion." and "This is wasteful, since it trains the model from scratch each time..." (and similar statement in Sec. 2.1): Those statements are quite debatable. There are lines of work, e.g., in Bayesian optimization, to model early stopping/learning curves (e.g., Domhan2014, Klein2017 and references therein) and where training procedures are explicitly resumed (e.g., Swersky2014, Li2016). The paper should reformulate its statements in the light of this literature.

-"Uncertainty could conceivably be incorporated into the hypernet...". This seems indeed an important point, but it does not appear as clear how to proceed (e.g., uncertainty on w_phi(lambda) which later needs to propagated to L_val); could the authors perhaps further elaborate?

-I am concerned about the rigor/correctness of Theorem 2.1; for instance, how is the continuity of the best-response exploited? Also, throughout the paper, the argmin is defined as if it was a singleton while in practice it is rather a set-valued mapping (except if there is a unique minimizer for L_train(., lambda), which is unlikely to be the case given the nature of the considered neural-net model). In the same vein, Jensen's inequality states that Expectation[g(X)] >= g(Expectation[X]) for some convex function g and random variable X; how does it precisely translate into the paper's setting (convexity, which function g, etc.)?

-Specify in Alg. 1 that "hyperopt" refers to a generic hyper-parameter procedure.

-More details should be provided to better understand Sec. 2.4. At the moment, it is difficult to figure out (and potentially reproduce) the model which is proposed.

-The training procedure in Sec. 4.2 seems quite ad hoc; how sensitive was the overall performance with respect to the optimization strategy? For instance, in 4.2 and 4.3, different optimization parameters are chosen.

-typo: "weight decay is applied the..." --> "weight decay is applied to the..."

-"a standard Bayesian optimization implementation from sklearn": Could more details be provided? (there does not seem to be implementation there http://scikit-learn.org/stable/model_selection.html to the best of my knowledge)

-The experimental set up looks a bit far-fetched and unrealistic: first scalar, than diagonal and finally matrix-weighted regularization schemes. While the first two may be used in practice, the third scheme is not used in practice to the best of my knowledge.

-typo: "fit a hypernet same dataset." --> "fit a hypernet on the same dataset."

-(Franceschi2017) could be added to the related work section.

*References*

(Domhan2014) Domhan, T.; Springenberg, T. & Hutter, F. Extrapolating learning curves of deep neural networks ICML 2014 AutoML Workshop, 2014

(Franceschi2017) Franceschi, L.; Donini, M.; Frasconi, P. & Pontil, M. Forward and Reverse Gradient-Based Hyperparameter Optimization preprint arXiv:1703.01785, 2017

(Klein2017) Klein, A.; Falkner, S.; Springenberg, J. T. & Hutter, F. Learning curve prediction with Bayesian neural networks International Conference on Learning Representations (ICLR), 2017, 17

(Li2016) Li, L.; Jamieson, K.; DeSalvo, G.; Rostamizadeh, A. & Talwalkar, A. Hyperband: A Novel Bandit-Based Approach to Hyperparameter Optimization preprint arXiv:1603.06560, 2016

(Swersky2014) Swersky, K.; Snoek, J. & Adams, R. P. Freeze-Thaw Bayesian Optimization preprint arXiv:1406.3896, 2014

*********
Update post rebuttal
*********

I acknowledge the fact that I read the rebuttal of the authors, whom I thank for their detailed answers.

My minor concerns have been clarified. Regarding the correctness of the proof, I am still unsure about the applicability of Jensen inequality; provided it is true, then it is important to see that the results seem to hold only for particular hyperparameters, namely regularization parameters (as explained in the new updated proof). This limitation should be exposed transparently upfront in the paper/abstract.
Together with the new experiments and comparisons, I have therefore updated my rating from 5 to 6.

---

> ### Comment · AnonReviewer2 · 2017-12-03
> **comment on citation**
>
> Franceschi 2017 should be cited as an ICML 2017 paper and not as a preprint

---

> ### Author Response · Authors · 2018-01-05
> **Response to Reviewer 1**
>
> Dear Reviewer 1,
>
> Thank you for your detailed comments. I will address your comments in the order in which you wrote them. For clarity, I have placed your comments in [brackets] and my responses will follow in plain text.
>
>
> [-"...and training the model to completion." and "This is wasteful, since it trains the model from scratch each time..." (and similar statement in Sec. 2.1): Those statements are quite debatable. There are lines of work, e.g., in Bayesian optimization, to model early stopping/learning curves (e.g., Domhan2014, Klein2017 and references therein) and where training procedures are explicitly resumed (e.g., Swersky2014, Li2016). The paper should reformulate its statements in the light of this literature.]
>
>
> Good point!  Fixed introduction to include lines of work exploring resumed training.
>
>
> [-"Uncertainty could conceivably be incorporated into the hypernet...". This seems indeed an important point, but it does not appear as clear how to proceed (e.g., uncertainty on w_phi(lambda) which later needs to propagated to L_val); could the authors perhaps further elaborate?]
>
> We believe using stochastic variational inference, as in the Bayes by Backprop paper may be leveraged to incorporate uncertainty into the hypernet.  This is now mentioned in the paper.
>
>
> [-I am concerned about the rigor/correctness of Theorem 2.1; for instance, how is the continuity of the best-response exploited? Also, throughout the paper, the argmin is defined as if it was a singleton while in practice it is rather a set-valued mapping (except if there is a unique minimizer for L_train(., lambda), which is unlikely to be the case given the nature of the considered neural-net model). In the same vein, Jensen's inequality states that Expectation[g(X)] >= g(Expectation[X]) for some convex function g and random variable X; how does it precisely translate into the paper's setting (convexity, which function g, etc.)?]
>
> Continuity of the best-response is exploited to guarantee a universal approximator can approximate the best-response.  It is a good point that the solutions are almost certainly set valued for \phi and w.  We now mention this, but leave notation as if it were a singleton for simplicity.  The convex function is min_{\phi} whose argument is a random variable that’s a function, L_\lambda(w_\phi) for \lambda in supp(p(\lambda)). L_\lambda(w_\phi) are sampled by sampling \lambda ~ p(\lambda) and currying it in L(w_\phi, \lambda).
>
>
>
> [-More details should be provided to better understand Sec. 2.4. At the moment, it is difficult to figure out (and potentially reproduce) the model which is proposed.]
>
>
> Added a line mentioning the simplest way to do 2.4 is by using a linear network, a randomly initialized current hyperparameter, and a conditional hyperparameter distribution that is a normal distribution centered on the current hyperparameter.
>
>
> [-The training procedure in Sec. 4.2 seems quite ad hoc; how sensitive was the overall performance with respect to the optimization strategy? For instance, in 4.2 and 4.3, different optimization parameters are chosen.]
>
>
> We have made the optimizer Adam for the hypernet and hyperparameter with identical optimizer parameters on all experiments.  The algorithm is easy to tune with Adam, but more sensitive when using SGD.
>
>
> [-"a standard Bayesian optimization implementation from sklearn": Could more details be provided? (there does not seem to be implementation there http://scikit-learn.org/stable/model_selection.html to the best of my knowledge)]
>
> We have added a link the gitHub of the Bayesian optimization implementation we use, but also moved all comparisons with Bayesian optimization to the appendix.
>
>
> [-The experimental set up looks a bit far-fetched and unrealistic: first scalar, than diagonal and finally matrix-weighted regularization schemes. While the first two may be used in practice, the third scheme is not used in practice to the best of my knowledge.]
>
> This is true - we wanted an experiment with an excessive number of hyperparameters that was defined in the related paper Maclaurin 2015

---

### Official Review · AnonReviewer2 · 2017-12-03
**Stochastic Hyperparameter Optimization through Hypernetworks**

**Rating:** 6
**Confidence:** 1

**Review:**

[Apologies for short review, I got called in late. Marking my review as "educated guess" since I didn't have time for a detailed review]

The authors model the function mapping hyperparameters to parameter values using a neural network. This is similar to the Bayesian optimization setting but with some advantages such as the ability to evaluate the function stochastically.

I find the approach to be interesting and the paper to be well written. However, i found theoretical results have unrealistic assumptions on the size of the network (i.e., rely on networks being universal approximator, whose number of parameters scale exponentially with the dimension) and as such are not more than a curiosity. Also, the authors compare their approach (Fig. 6) vs Bayesian optimization and random search, which are approaches that are know to perform extremely poorly on high dimensional datasets. Comparison with other gradient-based approaches (Maclaurin 2015, Pedregosa 2016, Franceschi 2017) is lacking.

---

> ### Author Response · Authors · 2018-01-05
> **Response to Reviewer 2**
>
> Dear Reviewer 2,
>
> Thank you for your comments. We have added a comparison to Maclaurin 2015, instead of Bayesian optimization/random search, showing that our algorithm reaches sub-optimal solutions faster than differentiating through unrolled optimization.

---

### Decision · Program_Chairs · 2018-01-29
**ICLR 2018 Conference Acceptance Decision**

**Decision:**

Reject

**Comment:**

The paper is interesting, and the update to the paper and additional experiments has already improved it in many ways, but the paper still does still not have as much impact as it could, by further strengthening the comparisons and usefulness in many of situations of current practice.